# Water chemistry reveals a significant decline in coral calcification rates in the southern Red Sea

Zvi Steiner [1], Alexandra V. Turchyn [1], Eyal Harpaz[2] & Jacob Silverman[3]

Experimental and field evidence support the assumption that global warming and ocean acidification is decreasing rates of calcification in the oceans. Local measurements of coral growth rates in reefs from various locations have suggested a decline of ~6–10% per decade since the late 1990's. Here, by measuring open water strontium-to-alkalinity ratios along the Red Sea, we show that the net contribution of hermatypic corals to the $CaCO_3$ budget of the southern and central Red Sea declined by ~100% between 1998 and 2015 and remained low between 2015 and 2018. Measured differences in total alkalinity of the Red Sea surface water indicate a $26 \pm 16\%$ decline in total $CaCO_3$ deposition rates along the basin. These findings suggest that coral reefs of the southern Red Sea are under severe stress and demonstrate the strength of geochemical measurements as cost-effective indicators for calcification trends on regional scales.

[1] Department of Earth Sciences, University of Cambridge, CB2 3EQ Cambridge, UK. [2] ZIM Integrated Shipping Ltd., Haifa, 31016, Israel. [3] Israel Oceanographic & Limnological Research, Haifa, 31080, Israel. Correspondence and requests for materials should be addressed to Z.S. (email: steinerz@gmail.com) or to J.S. (email: jacobs1@ocean.org.il)

Absorption of excess anthropogenic $CO_2$ from the atmosphere into the oceans is reducing seawater pH and thus the carbonate ion concentration, making it increasingly more difficult for calcareous organisms to build their skeletons[1,2]. The anticipated decrease in $CaCO_3$ production with increasing atmospheric $CO_2$ and resulting ocean acidification will significantly impact many aspects of the marine carbon cycle and lead to the deterioration of shallow-water-carbonate platform habitats, such as coral reefs[3,4]. The effect of ocean acidification may be particularly hard for coral reefs as corals, which form the foundation of their carbonate framework will not be able to produce $CaCO_3$ at a rate that will equal or offset the sum of mechanically and biologically mediated erosive processes[4].

The adverse effects of ocean acidification on hermatypic coral calcification, as well as whole coral reef community calcification has been demonstrated experimentally in numerous laboratory experiments[5] and two controlled field experiments[6,7]. These findings have motivated numerous studies of coral reef community metabolism, which established a baseline for our understanding of temporal variability in photosynthesis, respiration, calcification and $CaCO_3$ dissolution rates of whole reef communities[8–10]. For example, measurements of community calcification made over two annual cycles in the Eilat Nature Reserve Reef, northern Red Sea, have been used to develop a gross coral reef calcification rate equation, which is a function of live coral coverage, reef-water temperature and aragonite saturation[11]. Using this equation, together with modelled values of sea surface temperature and aragonite saturation for different future levels of atmosphere $CO_2$, it has been predicted that many tropical coral reefs might not be able to maintain their calcareous frameworks by the middle of the 21st century, when atmospheric $CO_2$ is expected to double relative to its pre-industrial level[4]. Studies of community metabolism in the Great Barrier Reef compared rates of calcification measured in the past few years with similar measurements conducted 3–4 decades ago, which together suggested an alarming decline in net calcification rates, and confirmed the predicted decline according to the Eilat rate equation[12,13]. In addition, numerous coral growth records derived from coral cores taken from live corals in the Great Barrier Reef, Red Sea and reefs in southeast Asia indicated that growth rates have been continuously declining since the 1990's, on the order of ~6–10% per decade[14–16]. In contrast, the same type of data suggests that coral calcification rates were stable in the decades preceding 1996 and 1998 in the Florida Keys and the Red Sea, respectively[14,17].

The alarming global decline in the state of coral reefs is largely due to periods of prolonged thermal stress that are increasing in frequency and duration, resulting in massive coral bleaching and mortality[18], in addition to local stress factors, such as coral mining and eutrophication[19,20]. This global decline in the state of coral reefs warrants careful monitoring of these important ecosystems. Assessment of the state of coral reefs has traditionally relied mostly on annual visual community structure surveys[21]. This method provides a wealth of information regarding the state of corals reef communities and allows the exploration of the processes that influence the state of the communities, yet the spatial coverage of these surveys is limited and they are very labour intensive. A different approach to assess the state of whole coral reef ecosystems is to measure changes in the water chemistry induced by biological activity, where precipitation of $CaCO_3$ by reef organisms (mainly hermatypic corals) induces changes in seawater total alkalinity ($A_T$)[22,23]. Furthermore, it has been proposed that this method can be applied to ocean basins or oceanic regions with a high prevalence of coral reefs, where changes in $A_T$, in conjunction with strontium and calcium concentrations that diverge from conservation with salinity can be used to determine the relative contributions of corals and calcareous plankton to their $CaCO_3$ budget[24]. In the oceans, $A_T$ is typically conservative with seawater salinity, which itself changes mostly due to evaporation and/or precipitation. Upward divergence from the oceanic conservation of $A_T$ relative to salinity in surface waters indicates net dissolution of $CaCO_3$ or independent production of $A_T$, while downward divergence of $A_T$ relative to salinity in surface waters indicates net $CaCO_3$ precipitation or independent uptake of $A_T$. Thus, an increase in the slope of $A_T$ vs. salinity relative to a baseline slope in a defined oceanic region could indicate a decrease in net calcification. If the $A_T$ vs. salinity slope is higher than the slope of oceanic conservation, then it would indicate that the system has become a net source of $A_T$; being a source of alkalinity could indicate the dissolution of $CaCO_3$ or an external supply of $A_T$, such as riverine or groundwater input[25–27].

The physical oceanography of the Red Sea features several characteristics which make it an ideal basin for ocean-chemistry-based exploration and monitoring of the $CaCO_3$ cycle. It is a long and narrow basin located in a hyper arid region with no significant river discharge or terrestrial runoff[28]. The only significant source of water to the Red Sea is surface water entering through the Straits of Bab-el-Mandeb, connecting the Red Sea with the Gulf of Aden and Indian Ocean[29,30]. The shallow sill at Bab-el-Mandeb prevents intermediate and deep Indian Ocean waters from entering the Red Sea[29]. Red Sea intermediate and deep waters form in the northern Red Sea and in the Gulfs of Aqaba and Suez, and are isolated from global ocean deep waters[30,31]. As a result, the Red Sea deep water mass has an unusually high temperature of 21 °C, maintaining supersaturation with respect to calcite and aragonite at all depths[32]. Upper thermocline water in the south and central Red Sea is warmer throughout the year and less salty than the deep and intermediate waters of the Red Sea, suppressing deep water formation and ventilation in these regions. The Red Sea is of particular interest in the context of ocean acidification and climate change since it is home to the world's longest continuous coral reef. The coral reefs of the southern Red Sea flourish under high summer time temperatures, which are generally above the thermal stress threshold considered to cause bleaching[33].

A Rayleigh distillation model has been previously used to calculate the contribution of coral reefs and calcareous plankton to carbonate production along the Red Sea, Gulf of Aden and north-western Indian Ocean based on data collected in October–November 1998[24]. These calculations provide a baseline for long-term monitoring of changes in calcification on a basin scale, which we are now able to compare with new data collected along the Red Sea during December 2015–January 2016, April 2016 and March 2018 (Fig. 1). Comparing these new and old data, we find that changes in the salinity-normalized concentrations of strontium and $A_T$ along the north-south bisecting axis of the Red Sea indicate a $26 \pm 16\%$ decline in net calcification rates between 1998 and 2015. This is a dramatic decrease for a region that has been previously declared a refuge for corals and coral reefs in an age of global warming and ocean acidification[3,33]. It seems that most of this decrease can be attributed to a reduction in net coral reef calcification rates in the tropical Red Sea while plankton calcification rates have also decreased. In addition, it is possible that dissolution in coral reefs increased as a result of coral bleaching, eutrophication and/or ocean acidification. Much of this region is presently inaccessible to researchers due to ongoing conflicts taking place in Yemen and Somalia, hence this large scale change could not be detected through standard visual surveys of coral reefs. This study demonstrates the strength of

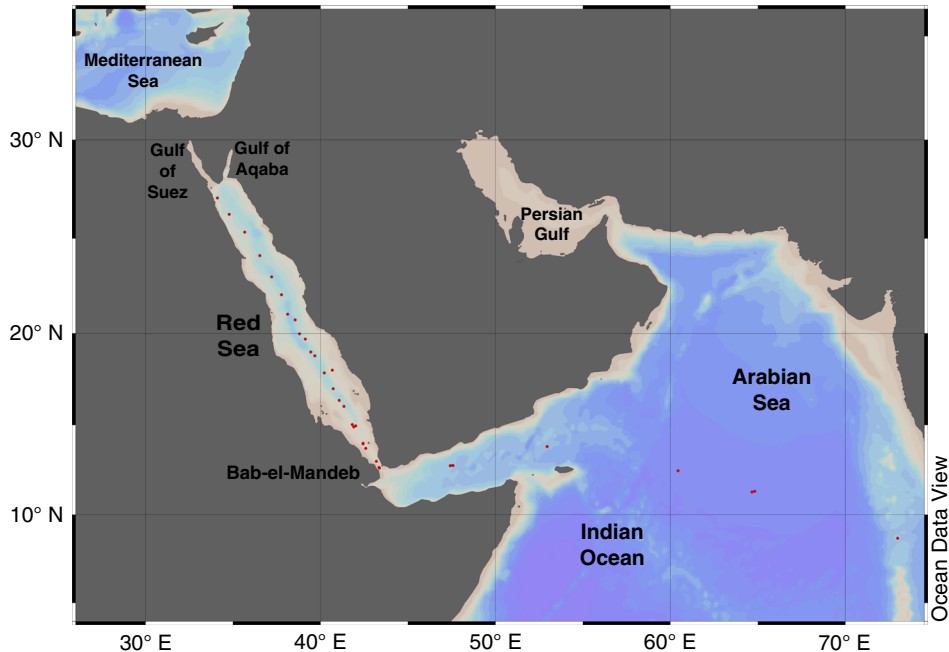

**Fig. 1** Location map of surface water samples collected for the present study. The map was created using Ocean Data View 5.0[79]. Supplementary Table 1 provides details of the exact sampling locations and times, and the physical and chemical parameters measured for the present work

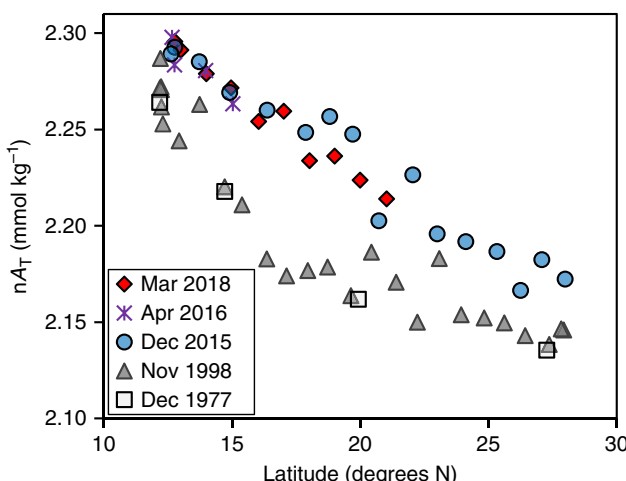

**Fig. 2** Total alkalinity normalized to salinity of 35 ($nA_T$) in the Red Sea and Gulf of Aden (West of 48°E). Analytical uncertainties in the present and previous studies[24,35] are smaller than the symbol sizes

geochemical tools in establishing baselines and following trends to determine environmental status on a large spatial and temporal scale.

## Results and Discussion

**Changes in the $CaCO_3$ budget of the Red Sea.** The Red Sea region as a whole is a significant producer of $CaCO_3$, and the bulk of this $CaCO_3$ is not subsequently dissolved; as a result, the Red Sea region is a net sink for $CaCO_3$. We can see this net sink of $CaCO_3$ in the gradual decrease in salinity-normalized total alkalinity ($nA_T$) in the Red Sea surface water as it flows northward from the straits of Bab-el-Mandeb (Fig. 2). The $nA_T$ of the Red Sea deep water is lower than surface water $nA_T$ at the site of deep water formation, an observation that has been attributed to inorganic precipitation of $CaCO_3$ on the surfaces of dust grains and re-suspended or deposited sediment grains[34]. Comparison of

data collected during 2015–2018 with data collected in previous cruises along the Red Sea[24,35] show that uptake of $A_T$ along the Red Sea transect decreased in 2015 relative to 1998 (Fig. 2). The $nA_T$ of surface water samples collected in April 2016 and March 2018 is similar to the December 2015 data. The spatial distribution of this uptake also changed, $A_T$ uptake decreased significantly in the southern Red Sea and western Gulf of Aden since 1998 and possibly increased from 20°N northward (Fig. 2), as the gap between the data sets becomes smaller in that region. The uptake of $A_T$ in the Red Sea is equivalent to the difference between the slope of $A_T$-to-salinity in the Red Sea and the Indian Ocean (IO) surface waters. In 1998, the slope of $A_T$-to-salinity between the IO and Red Sea was $26.6 \pm 2.0$, while in 2015 and 2018 the slope was $35.8 \pm 2.4$ (Fig. 3). Assuming that the residence time of water in the Red Sea did not change significantly between 1998 and 2015, the decrease in $A_T$ uptake in 2015 relative to 1998 is $26 \pm 16\%$. The Indian Ocean $A_T$-to-salinity slope in 1998 and 2015–2018 remained unchanged with a value of $61.9 \pm 4.0$. The residence time of surface waters in the Red Sea is on the order of 1 year[36], hence each of the profiles presented in Figs. 2 and 3 represents a yearly cycle and should be sensitive to interannual variability.

The full data set suggests that the 2015 surface water data is reliable for calculations of calcification rates in the central and southern Red Sea but not in the northern Red Sea. Concentrations of calcium and $nA_T$ ($\Delta Ca^{+2}$: $\Delta nA_T$) decrease at a ratio of $0.55 \pm 0.11$ mole mole$^{-1}$ along the southern and central Red Sea, in agreement with predicted ratios in precipitation of $CaCO_3$. However, in the northern Red Sea (north of 24.1°N), roughly one mole of calcium is added per two moles of $nA_T$ lost ($\Delta Ca^{+2}$: $\Delta nA_T = -0.53 \pm 0.08$ mole mole$^{-1}$). Such ratios are characteristic of hydrothermal activity and reactions between basaltic minerals and seawater[37]. Mineralogical transformations of fresh basalts and mixing of seawater with hydrothermal fluids replace calcium by magnesium, acting as a net source of dissolved calcium and sink of dissolved magnesium for the overlying seawater[38–40]. Strontium concentrations, on the other hand, are typically not significantly altered relative to chlorinity during high temperature circulation[41]. It was previously reported that surface water from

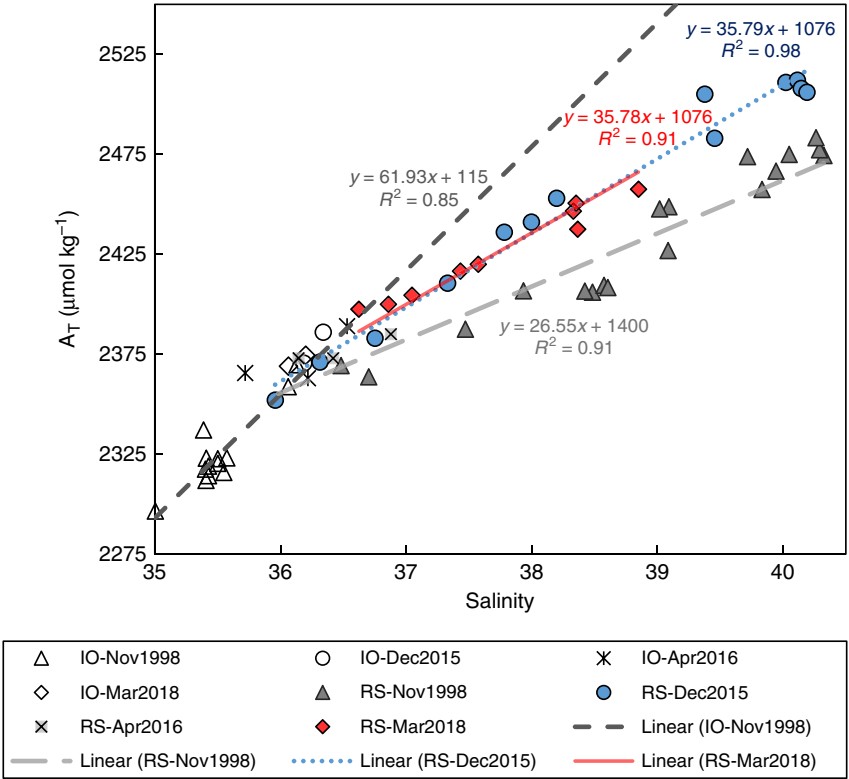

**Fig. 3** Surface water total alkalinity versus salinity. The figure includes all surface water samples collected from the Indian Ocean (IO), Arabian Sea, Gulf of Aden and Red Sea (RS) in November 1998, December 2015, April 2016 and March 2018. Analytical uncertainties are smaller than symbol sizes

this region is slightly depleted in magnesium, an observation which further supports that hydrothermal/basalt alteration reactions are important[24]. The presence of hot brines in the deepest parts of the Red Sea[40], as well as the overall relative narrowness of the basin, suggests that both sources are possible contributors of calcium. Unlike the northern Red Sea, mixing of brine waters into surface waters of the central and southern parts of the Red Sea is of minor importance due to the development of a steep density gradient between the surface and deep waters.

**Relative contributions of corals and calcareous plankton**. Each group of calcifying organisms precipitates $CaCO_3$ with a typical tendency to incorporate strontium in the mineral lattice[42] ($K_D$, the distribution coefficient, is the ratio between a trace element and the major element in a solid versus the fluid it precipitates from). Corals precipitate aragonite with a slight preference for strontium over calcium ($K_D^{reef} = 1.04 \pm 0.03$[43]). The distribution coefficient of strontium in aragonite precipitated by red and green algae as well as inorganic aragonite are also high[42,44]. Common calcareous plankton, on the other hand, typically precipitate strontium-poor $CaCO_3$. For example, planktonic foraminifera precipitate low Mg-calcite with typical strontium distribution coefficient of $K_D = 0.12$, while pteropods precipitate aragonite with $K_D = 0.12$, and coccolithophorids precipitate low Mg-calcite with a typical $K_D \approx 0.3$[42,45,46]. Taking all these calcifying plankton together, we can estimate an "average" distribution coefficient of strontium in planktonic $CaCO_3$ precipitated in the Red Sea, $K_D^{plankton}$, is $0.17 \pm 0.03$, similar to the average value for biotic low magnesium calcite in general[47]. Because this number is significantly different from the distribution coefficient for strontium precipitated in corals ($K_D^{reef} = 1.04 \pm 0.03$), by measuring the variations in strontium, calcium and alkalinity in the seawater along a transect we can determine the relative

contributions of corals and calcareous plankton to the $CaCO_3$ budget of a defined oceanic region.

The average distribution coefficient of strontium in all $CaCO_3$ precipitated along a surface water flow trajectory can be calculated using the Rayleigh distillation equation[24]

$$\frac{R_w}{R_0} = f^{K_D - 1} \qquad (1)$$

where $R_w$ is the Sr/Ca or Sr/$A_T$ ratio in each surface seawater sample; $R_0$ is the Sr/Ca or Sr/$A_T$ ratio in a reference sample and $f$ is the fraction of salinity normalized calcium or $A_T$ lost relative to the reference sample.

In the present work, we focus on changes in $A_T$ relative to salinity and Sr/$A_T$ ratios since the relative changes in the concentrations of dissolved calcium were very small. If salinity normalized concentrations of calcium, $A_T$ and strontium change only due to precipitation of $CaCO_3$, it is reasonable to assume a two-end-member system in which one end member is coral aragonite and the other is calcareous plankton, each with a typical distribution coefficient for strontium in its precipitated carbonate (much lower for calcareous plankton than for coral aragonite). Under this assumption, the relative net contributions of plankton ($X_{plankton}$, which includes benthic precipitators of calcite) and corals ($1 - X_{plankton}$, which includes inorganic aragonite) to the $CaCO_3$ budget in the Red Sea can be expressed by Eq. (1) and solved using a water mass chemical balance[24]:

$$K_D = K_D^{plankton} \cdot X_{plankton} + K_D^{reef} \cdot \left(1 - X_{plankton}\right) \qquad (2)$$

An important underlying assumption of the Rayleigh distillation model is that there is no back reaction[48], i.e. no dissolution of $CaCO_3$. Thus, Eq. (1) is not typically suitable for assessments of the $CaCO_3$ cycle in deeper water masses or in shallow water

environments where inorganic or biogenic dissolution, respectively may play an important role in their $CaCO_3$ budget. In the Red Sea, deep water $CaCO_3$ dissolution is not prevalent because of the high temperature and saturation states of $CaCO_3$ minerals. In-situ dissolution of $CaCO_3$ at the site of its formation, as part of the diurnal cycle, does not measurably modify the chemistry of the water far away from the reef as long as dissolution is balanced by accretion, hence this natural process is normally transparent in our calculations. An additional underlying assumption of the model is that all calcification processes occurring within a body of water are reflected by its chemical composition, i.e. laterally mixed. The Red Sea has a channel like structure and a single significant entry point, ensuring that nearly all water found in the surface layer of the northern Red Sea originated from the strait of Bab-el-Mandeb, connecting the Red Sea with the Gulf of Aden and the Indian Ocean. Within this channel, a series of mesoscale eddies vigorously mixes the upper water column on the longitudinal axis[30], as reflected by the zonation of many physical and biological parameters[49].

The average distribution coefficient of strontium in the precipitated $CaCO_3$, as calculated using the Rayleigh distillation model for the south and central Red Sea, declined from $0.36 \pm 0.20$ in 1998 to $0.11 \pm 0.06$ in 2015 and $0.11 \pm 0.07$ in March 2018 (Fig. 4). The earlier value has large margins of uncertainty when calculated based on $Sr/A_T$ ratios alone, yet it is in excellent agreement with calculations based on calcium concentrations as well as long term records of coral and plankton calcification rates in this region, supporting its accuracy[24]. In terms of relative contributions, the 2015 and 2018 average distribution coefficients for the Red Sea suggest that the current regional $CaCO_3$ cycle is dominated by planktonic foraminifera and pteropods, while the earlier average $K_D$ suggests that ~20% of the $CaCO_3$ was precipitated in coral reefs along with possible contributions from coccolithophores[24]. These water-chemistry based calculations are

supported by reports of severe coral bleaching along the central Red Sea coast of Saudi Arabia in 2015[18,50] and suggest that the extent of this bleaching event included the southern Red Sea. Our calculations also suggest that calcification rates of the southern and central Red Sea corals remain low and that the decrease represents a prolonged episode rather than a short term event isolated in 2015. By subtraction, the decrease in total calcification rates and coral calcification rates of the southern and central Red Sea suggests that calcification rates of foraminifera and pteropods decreased by $7.5 \pm 7.5\%$. It should be noted that in any case this decrease in $K_D$ does not mean that all corals are dead in this region but rather indicates an overall equilibrium between $CaCO_3$ production and dissolution in coral reefs and shallow carbonate sediments derived from coral reef erosion. Any further decline in reef calcification rates or increase in dissolution rates should result in net loss of $CaCO_3$ from the reef framework and ultimately the degradation and loss of this important habitat.

A difficulty that has to be taken into account when considering the validity of the Rayleigh distillation model in assessments of contributions to the $CaCO_3$ cycle is that strontium can be removed/supplied by additional sources. Hydrothermal activity does not seem to be significant in this respect since reactions between seawater and basaltic minerals act in opposite direction to mixing hydrothermal fluids with seawater, cancelling the effects of each other[41]. There are also no significant rivers flowing into the Red Sea. Acantharia, which are radiolarian precipitating $SrSO_4$ skeletons may be abundant in the surface mixed layer throughout the region[51]. Precipitation of $SrSO_4$ is expected to remove large amounts of strontium from the water, increasing the apparent distribution coefficient of strontium, while dissolution of even small amounts of acantharia may decrease the apparent distribution coefficient to below zero. Given that seawater is highly undersaturated with respect to $SrSO_4$[52], its imprint on profiles of strontium concentrations depends on the export efficiency of the shells of these plankton. It was previously shown that dissolution rates of silicate radiolarians in the undersaturated water column increase exponentially with increasing temperatures[53] and it is likely that the same is true for the far more soluble acantharia shells. Absence of apparent contribution by precipitation/dissolution of acantharia shells to the Sr/Ca ratios of the Red Sea in the southern and central Red Sea data, suggest that the vast majority of the acantharia dissolve within the warm surface mixed layer of this region. For the 1998 data set it was shown that there is excellent agreement between calculated precipitation rates of corals and calcareous plankton with satellite and sediment core data from this region, suggesting that the relative contribution of acantharia was not significant for our budget calculations[24].

**Long term trends in the Gulf of Aqaba**. The northernmost Gulf of Aqaba is the only region in the Red Sea system in which long-term monitoring of the ecology and chemistry of the marine environment has been conducted routinely since the 1990's. Located at the terminus of the Red Sea surface water flow, water chemistry in the Gulf of Aqaba sums the processes occurring along the Red Sea with a likely overrepresentation of local processes occurring within the Gulf. In the northern Gulf of Aqaba, coral cover and the density of colonies declined significantly during the 1990's and reached an alarming low in 2004[54]. This decline was considered to be a result of eutrophication due to nutrient emissions from nearby fish farm activity[8,54,55]. The state of the coral reef in Eilat improved after 2004, following reductions in fish production and ultimate removal of this nutrient source between 2006 and 2008[56,57]. Changes in the abundance and species composition of planktonic foraminifera in recent decades

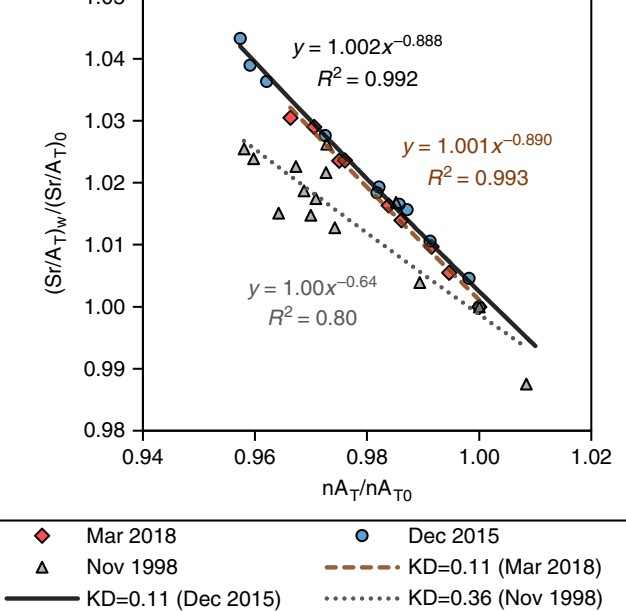

**Fig. 4** Calculation of the average distribution coefficient of strontium in $CaCO_3$ precipitated along the southern and central Red Sea (Bab-el-Mandeb to 24°07′N). Data collected in 1998 is from ref. [24]. Power of the trendline equals the distribution coefficient minus one (Eq. (1)). The average distribution coefficient of strontium in $CaCO_3$ was therefore $0.36 \pm 0.16$ in 1998, $0.11 \pm 0.06$ in 2015 and $0.11 \pm 0.07$ in 2018 with 95% confidence intervals

have also been reported in the Gulf of Aqaba[58]. Monthly measurements of $A_T$ in the northern Gulf of Aqaba suggest a decline in calcification rates between 1999 and 2004[59] and relative long-term stability with seasonal and inter-annual variability since 2008[34,56]. Recorded increase in $A_T$ of the surface waters of the Gulf of Aqaba in the years following 1998, probably reflects the local decline in the state of coral reefs in the northern Gulf of Aqaba[54] as well as decreased calcification rates along the main basin of the Red Sea[14,60].

**Factors contributing to reduced calcification rates**. Three main processes are most likely to contribute to reduced calcification rates along the southern Red Sea: Decreased carbonate ion concentration and aragonite saturation due to absorption of anthropogenic $CO_2$ by the surface waters[61]. Increased temperatures in this already very warm region[14,50,62,63]. Eutrophication, which may support colonization of hard substrates by macro algae and displacement of reef building corals, as well as the increase in boring activity (biogenic dissolution of $CaCO_3$) by organisms that inhabit the $CaCO_3$ framework of coral reefs[19,64] and the bacterially mediated dissolution of carbonate sediments, which is also affected by ocean acidification[20,65,66]. In the following discussion, we shall see that all three processes probably contribute to the observed decline.

Rates of coral calcification strongly depend on the saturation state of aragonite[11]. The same is probably also true for calcite precipitation by foraminifera[67]. The partial pressure of $CO_2$ in surface waters of the Red Sea have been shown to be near equilibrium with atmospheric $CO_2$[68], hence the ecosystem should quickly respond to changes in atmospheric $CO_2$. At the temperature-salinity range in the study area, increased temperatures increase the aragonite saturation state by 0.03 units-per-degree Celsius and may either increase or decrease biotic calcification rates, depending on the physiological response of organisms to temperatures. A null hypothesis of no change in live coral cover suggests that the increase in $pCO_2$ and warming of 0.25 °C[63] between 1998 and 2015 should have decreased net coral calcification rates along the Red Sea by 4.4%, based on the observed response of a coral reef community in the Gulf of Aqaba to seasonal changes in the saturation state of aragonite and temperature (ref. [11]; Supplementary Fig. 1). Therefore, the dramatic decline in net coral calcification rates in the southern and central Red Sea as inferred from Figs. 2 to 4 must result from additional processes.

Reduced rates of coral calcification and reduced abundance of large corals since 1998 has been previously reported for several Red Sea reefs[14,60]. It was suggested that this decline in coral growth rates was a result of a long series of warm years in the central Red Sea[14]. In 2015, global temperatures were particularly high and widespread coral bleaching events were reported globally, including sites in the central Red Sea, Persian Gulf and Indian Ocean[18,50,62]. In the Red Sea, sea surface temperatures in 2015 were 0.5 °C higher than the long term average, yet while 2015 was warmer than average for this region, it was not the warmest on record in recent years[63]. The bleaching event clearly affected the metabolic state of Red Sea corals yet coral bleaching is a very rapid process, which is unlikely to induce a gradual decline in coral growth as documented for the central Red Sea[14]. It therefore seems that while bleaching played a major role in decreasing coral calcification rates, change in calcification rates started earlier. Data from April 2016 and March 2018 show that recovery from the 2015 bleaching event has yet to happen.

A third possible contributor to the observed decline in southern Red Sea net calcification rates is increased erosion. Increased nutrient availability and subsequent growth of epilithic algae, increases the activity of boring organism and dissolution rates of $CaCO_3$ in coral reefs[69,70] as well as bacterially mediated dissolution of carbonate sediments and rubble[20]. For example, between 1960–1970 and 2009, the night-time dissolution rate of a reef in One Tree Island increased from 31 to 71% of the gross calcification rates, a three-fold increase in absolute terms[13], likely driven by warming and ocean acidification.

Much like the Gulf of Aqaba, the central and southern parts of the Red Sea are increasingly threatened by overfishing, urban development and eutrophication[71]. The effect of eutrophication on trophic levels in central Red Sea coral reefs has been clearly documented[72] and should impact the whole community growth and calcification rates. Support for this claim comes from the average regional distribution coefficient of strontium, suggesting that coccolithophores calcification rates have also decreased. Calcification rates by coccolithophores are less likely to respond dramatically to small changes in temperature or $pCO_2$ and more likely to be affected by eutrophication and pollution as these give rise to increased abundance of faster growing species[73,74].

## Methods
**Sampling**. During 27/12/2015–3/1/2016, 18–21/4/2016 and 23–31/03/2018 seawater samples were collected from the sea surface by bucket off the deck of the Container Ships ZIM Qingdao and Yokahama during their passage northward from the Arabian Sea through the Gulf of Aden and the Red Sea on their way to the Mediterranean (Fig. 1). Water samples were kept in refrigeration in 1.5 L gas tight plastic bottles until they were subsampled at the Israel Oceanographic and Limnological Research (IOLR) lab in Haifa, Israel within 3 days from collection of the last sample. Water samples for total alkalinity ($A_T$), dissolved inorganic carbon (DIC) and density were kept in brown glass bottles and samples for Sr, Ca and Na analysis were kept in polypropylene tubes. Samples for $A_T$, DIC and density were measured within the first week after arriving at IOLR.

**Carbonate chemistry**. Total alkalinity was determined at IOLR by potentiometric Gran titration of ~22 g subsamples, filtered through Whatman GFF 0.45 µm filters using a Metrohm Titrino 785 Titrameter with a temperature corrected pH probe and HCl 0.05 N. $A_T$ was calculated from the intercepts and slopes of the linear regression fits between measured pHs and corresponding acid volumes in the pH range 3.8 and 3.3[75]. The acid concentration was calibrated using seawater CRMs from A. Dickson's lab. The precision of these measurements was ± 2 µmol kg$^{-1}$ (two measurements per sample). The DIC was extracted from 1.6 mL sub-samples by acidifying them with phosphoric acid ($H_3PO_4$, 10%) using an automated $CO_2$ extractor and delivery system (AERICA by MARIANDA) and high grade $N_2$ (99.999%) as a carrier gas connected on line with a LiCor 6252 IR $CO_2$ analyser. Measurements were calibrated using seawater Dickson CRMs. The repeatability of the measurements was 1.7 ± 1.3 µmol kg$^{-1}$ (mean ± STD of all measurement errors, $n = 28$).

**Salinity**. Salinity was calculated from density measurements conducted at IOLR using an Anton-Paar density and sound velocity metre, model DSA 5000 M. Together with the recorded temperature at the time of the measurements the results were input into the international equation of state for seawater[76,77]. The roots of the resulting polynomial with salinity as the unknown variable were calculated using the Newton–Raphson method. Validation of this method was done with a Dickson CRM and IAPSO standard seawater producing an accuracy of better than ±0.003.

**Major elements**. Concentrations of calcium, strontium and sodium were measured at the University of Cambridge by inductively coupled plasma optical emission spectroscopy using an Agilent Technologies 5100 ICP-OES. Samples were diluted with 0.1 N $HNO_3$ at 1:71 ratio and analysed in duplicates by sample standard bracketing. Initial calibration lines were obtained by running different dilutions of IAPSO seawater batch P157. For accuracy calibrations, IAPSO seawater was spiked with a $^{42}Ca$–$^{48}Ca$ spike; calcium was separated by ion chromatography by running the samples through Sr spec resin followed by AG50W-X8 resin and measured by Thermo Scientific Triton Plus thermal ionization mass spectrometry (TIMS). Sr/Ca ratios and the concentrations of strontium were verified by running a standard with a known Sr/Ca ratio, prepared by Mervyn Greaves and utilized in many previous studies at the laboratory of Harry Elderfield. For this study, we used the Ca422.673, Sr421.552 and Na468.821 spectral lines. $1\sigma$ standard deviation of the analyses was 0.12% and 0.11% for Ca/Na and Sr/Na ratios, respectively ($n = 33$). All calcium and strontium measurements were normalized to salinity of 35 by assuming sodium concentrations of 468.5 mmol kg$^{-1}$ at that salinity[78].

## Data availability

All data supporting the findings of this study is provided in the online Supplementary Information.

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

## Acknowledgements

This study was supported by ERC Grant Stg (307582) CARBONSINK to A.V.T. and a Blavatnik postdoctoral fellowship to Z.S. We thank Mervyn Greaves for advice and support in ICP analyses.

## Author contributions

J.S and Z.S. designed the study. E.H., J.S. and Z.S. performed research. Z.S., J.S. and A.V.T. analysed the data and wrote the paper.

## Additional information

**Competing interests:** The authors declare no competing interests.

