## [Peer Review File · Nature Communications]

Reviewers' comments:

Reviewer #1 (Remarks to the Author):

The paper describes a novel geochemical method for seeing trends in benthic and pelagic calcification at the basin-scale in the Red Sea. The data show that coral calcification has declined in recent decades by approximately 20% and pelagic calcification by approximately 7%. The interpretation that dissolution may have increased more than calcification has decreased is in agreement with other recent studies that acidification and eutrophication may have greater effects on dissolution than calcification. This study is important because it shows convincingly at the basin-scale that calcification rates of corals and to a lesser extent planktonic calcifiers have declined measurably in the last 17 years by a method that is completely independent of biological census methods. I find that the authors' interpretation of the chemical data is correct and supports the conclusions that they have made. This is an important paper and should be accepted.

A few minor quibbles with the text:

Line 56 Should cite Helmle et al 2011 that reported the calcification rates of corals in Florida have not changed significantly between 1936 and 1996. So not all reefs around the world are reporting declines even though this region has experienced severe bleaching events and the same amount of ocean acidification.

Line 169 It would be more accurate to say that calcification rates in the north (>20N) decreased less than further south given that nTA in 2015 is still greater than nTA in 1998 but to a smaller degree than further south.

Reviewer #2 (Remarks to the Author):

Given the timeliness and importance of reef monitoring, this is an interesting paper that draws some important conclusions concerning coral calcification in a particularly vulnerable point in time. It is important to be able to monitor reefs on a much more global scale without having to rely on ecological surveys which are expensive and time consuming.

I am concerned, however, about the overall conclusion that compares only 2 years on record (1998 and 2015) and attempts to make a claim for a decrease in coral calcification through time based on this. Two years do not make a trend, particularly when the most recent was one of the worst global coral years on record. One would expect the numbers derived in this year to be particularly skewed. I understand that monitoring through time has supported the overall conclusions drawn by the author, but this weakness needs to be addressed. The paper also lacks statistics, as p-values (or other such values) are not provided to report significance, etc of the calculations made. It would be helpful if some measure of the significance between numbers could be given (for example, when comparing years) such that the authors' confidence in their estimates could be addressed.

I agree that the system in question is interesting, particularly given its unique water circulation patterns and residence time of 1 year. This makes the system particularly useful for making observations of the immediate coral response to these stress events. However, it also highlights that any chemical observations made will record not the long-term averages but only something that has happened within a year. Given the lack of years in the dataset, this issue needs to be addressed.

Minor comments:

Line 59: Citation needed.

Lin 109: How do these other types of algae and inorganic precipitation impact the values reported later in the paper? They are mentioned here and never brought up again.

Line 140: Your discussion talks extensively about dissolution, which is assumed here to remain constant. Please revise or clarify.

Line 167: Sentence is very confusing. Please revise.

Line 222: In which direction are they temperature sensitive? This is very important for the argument, particularly given that temperature played such a large role in this particular event.

Line 263: To what degree does this change increase the saturation state? What were the actual temperature differences between 1998 and 2015? It seems this is extremely important for the claims made below. If warming was greater, wouldn't this have a larger impact on saturation? Bleaching? Was this warming the actual observed differences during one of the worst bleaching years on record? Generally, more reporting of actual measured values would be helpful.

266: Many different claims are made about saturation state, but saturation was not, to my knowledge, directly measured. A brief explanation of how these numbers were derived specifically for this paper would be helpful, or if they were not measured, a comment on how saturation may have differed between the values measured in the previous paper and the current one?

267: It was stated previously that there were large changes in coral cover due to the bleaching. Please clarify.

Line 292-295: This sentence is very confusing. Please revise and clarify. As written, the logic is hard to follow and put into context with the argument and previous values reported.

Line 330: These are not standard operating procedures for carbonate system parameters.

Figure 3: This figure, with the labeling as is, is very hard to read. For the color blind, it is nearly impossible to read. Please consider changing the figure to make the points clearer and the overall conclusions drawn more visible.

Reviewer #3 (Remarks to the Author):

General

The authors present an interesting data set that demonstrates the change in Sr/TA in the Central and Red Sea 1998 and 2015. This ratio can be used to distinguish between coral reef and open ocean planktonic precipitation of CaCO_3 by interpreting it using a Rayleigh distillation model. They observed a significant decline in KD, thus a decline of the contribution of coral reef calcification to the overall budget. In fact, the model revealed there was almost 0 contribution from coral reefs, and the basin-scale budget was entirely due to planktonic calcification. They also observed a decline in total calcification of ~26% based on salinity normalized TA (assuming no changes in circulation, residence time, etc). Since coral reefs contributed ~20 % of the budget in 1998, they also concluded that there was ~7% decline in planktonic calcification during this time. While I think this is an interesting data set, I think the authors have over interpreted their data and drew conclusions that are not supported by the data. Therefore I cannot recommend publication. I have outlined my objections below.

The main objection I have for this paper, is that the authors present two data points, one in 1998 and another in 2015, and attribute the changes in NCC to changes in environmental conditions (temperature, CO_2 , and eutrophication). This is problematic for many reasons. First, they do not consider any interannual variability. It would not be surprising to me if planktonic calcification rates exhibited interannual variability of ~7 %, which is the signal they observed here. Second, unfortunately 2015 was the year where coral reefs experienced mass bleaching events worldwide, including the Red Sea. A significant decline in NCC would be expected associated with this event. However, the authors make no attempt to quantify how this bleaching event may have contributed to the decline in the observed coral reef NCC. It is plausible that the signal was dominated by this event.

The Eilat reef equations assumes that there is no change in community composition, however, this is not a valid assumption. The authors mention that there has been documented declines in coral cover in this region. In addition, there was a massive global bleaching event in 2015 that affected reefs worldwide, including the Red Sea. Therefore it is hard to believe any inference based on this equation, and I recommend it to be removed from the manuscript.

The authors mention that the Rayleigh distillation model does not work if dissolution is a significant process, however, later goes on to argue that dissolution rates in coral reefs increased by 75% (from 20%). They do not mention how this would affect the model calculation (I assume another end member needs to be added for reef dissolution?) An uncertainty analysis for this seems warranted.

Finally, how can you be confident that the Central and Southern part of the Red Sea is unaffected by deep sea processes? If it is altered in the North, I would assume it would also have an influence in the Central and Southern parts as well. If deep waters have a different Sr/TA, Sr/Ca ratio, then this could affect the interpretation.

Specific

L38: A follow up paper to this experiment has now been published in Nature. So there are now 2 controlled field experiments.

L146-149: A single entry point into the Red Sea is not a good justification of a laterally mixed system.

L266: This assumption is not true, due to the changing coral community mentioned by the authors throughout the 2000's, and the bleaching event in 2015.

L274-L281: It is not clear to me what the authors are arguing here.

L292-L297: These calculations are dubious. See comments above.

L307: the strontium budget suggests that their calcification rates may have decreased, but doesn't necessarily mean their abundance has decreased.

L316: How do you get footprint of the chemical data without information of circulation patterns?

L330: Were the samples poisoned? It sounds like the samples could have sat around for 2 months before they were analyzed. If it wasn't fixed properly, then TA could get altered.

L343: a commercially available system isn't exactly 'custom'...

Figure 3: Is this all of the data? Or just south or 20 N?

Response to reviewers

Please find below our detailed response to the reviewers' comments in the order they appeared. For your convenience, we itemized the reviewers' comments in bold font followed by our action/ answer in normal font.

Reviewer #1:

Line 56 Should cite Helmle *et al* 2011 that reported the calcification rates of corals in Florida have not changed significantly between 1936 and 1996. So not all reefs around the world are reporting declines even though this region has experienced severe bleaching events and the same amount of ocean acidification.

Thank you for this observation. The reference to Holmle et al. was added along with the sentence "In contrast, the same type of data suggests that coral calcification rates were stable in the decades preceding 1996 and 1998 in the Florida Keys and the Red Sea, respectively (14, 17)."

Line 169 It would be more accurate to say that calcification rates in the north (>20N) decreased less than further south given that nTA in 2015 is still greater than nTA in 1998 but to a smaller degree than further south.

The reviewer is correct, normalized total alkalinity north of 20°N was indeed higher in 2015 than 1998. If we assume that the source of the northern surface water is surface water flowing from the south, it appears that the initial alkalinity of the northern waters in 2015 was higher than the initial alkalinity in 1998. The rate of decrease in total alkalinity from 20°N northward also appears higher in the 2015 data (the gap between the data sets shrinks), suggesting that calcification rates increased rather than decreased in the northern part of the basin. The sentence was slightly reworded "The spatial distribution of this uptake also changed, A_T uptake decreased significantly in the southern Red Sea and western Gulf of Aden since 1998 and possibly increased from 20°N northward (Fig. 2), as the gap between the data sets becomes smaller in that region."

Reviewer #2:

I am concerned, however, about the overall conclusion that compares only 2 years on record (1998 and 2015) and attempts to make a claim for a decrease in coral calcification through time based on this. Two years do not make a trend, particularly when the most recent was one of the worst global coral years on record. One would expect the numbers derived in this year to be particularly skewed. I understand that monitoring through time has supported the overall conclusions drawn by the author, but this weakness needs to be addressed.

The revised version of the manuscript includes a new sampling campaign from March 2018. The trend arising from these samples is very similar to the December 2015 data as well as to the April 2016 data, which appeared in the original supporting information but was not plotted in the earlier version of the paper (see revised figure 2 below). These data support the conclusion that the Red Sea system has reached a new state, where coral reefs in the southern and central parts of the basin have not recovered yet from the bleaching event or other stress factors.

The paper also lacks statistics, as p-values (or other such values) are not provided to report significance, etc of the calculations made. It would be helpful if some measure of the significance between numbers could be given (for example, when comparing years) such that the authors' confidence in their estimates could be addressed.

Thank you to the reviewer for this comment. The paper reports the 95% confidence intervals and coefficients of determination for each of the calculations presented or discussed. p-values are the statistical measure of the level of independence between two or more groups. Every calculation presented in the paper assumes that the chemical composition of water in the Indian Ocean and Gulf of Aden did not change measurably between 1998 and 1977, an assumption that we tested in the field. All calculations of Red Sea calcification rates converge from the single point of entry at the straits of Bab-el-Mandeb and are therefore defined as dependant groups of variables. As such, we feel that statistical tests of this type are not suitable for the present case.

I agree that the system in question is interesting, particularly given its unique water circulation patters and residence time of 1 year. This makes the system particularly useful for making observations of the immediate coral response to these stress events. However, it also highlights that any chemical observations made will record not the long-term averages but only something that has happened within a year. Given the lack of years in the dataset, this issue needs to be addressed.

Thank you for the comment. The original submission included a comparison of the 1998 data with the GEOSECS data collected in 1977 to show that the 1998 data represents the continuation of a long term trend. The paper by Steiner et al., 2014, which published the 1998

data, includes additional comparisons with data from 1964 and 1982, showing the same long-term trend. In the revised version of this manuscript, we compare the 1998 and 2015 data with newer data from April 2016 and March 2018. To fill in the gaps in data between 1998 and 2015, the manuscript discusses findings from a long term monitoring station located in Eilat, at the northern tip of the Red Sea and coral growth records from the central Red Sea showing that coral growth rates were stable during 1930-1998 and declined afterwards. Interestingly, all these surveys show that calcification rates in the Red Sea system were very stable until 1998, decreased significantly, and reached a new, possible stable state for at least the last three years.

Minor comments:

Line 59: Citation needed.

Thank you for this observation. References to Hughes et al., 2018, Chazottes et al., 2002 and Islam et al., 2016 were added to the sentence.

Line 109: How do these other types of algae and inorganic precipitation impact the values reported later in the paper? They are mentioned here and never brought up again.

The method used in this manuscript is not sensitive enough to distinguish among various types of aragonite except pteropode shells since they all have similar Sr/Ca ratios. The calculated relative contribution of corals is therefore not a pure end member but includes inorganic precipitation of aragonite and aragonite precipitated by calcareous algae. This point is made clear 20 lines later in the text with the explanation of the two-end member assumption “Under this assumption, the relative net contributions of plankton (X_{plankton} , which includes benthic precipitators of calcite) and corals ($1-X_{\text{plankton}}$, which includes inorganic aragonite) to the CaCO_3 budget in the Red Sea can be expressed by Eq. 1”. In the studied system, most, if not all inorganic aragonite precipitated is precipitating in dead parts of coral colonies (a process shown by Wurgaft et al., 2016). This inorganic aragonite therefore contributes to strengthening coral colonies, and is an integral part of the CaCO_3 fraction that should be considered as coral precipitation, even though it is formed by an abiotic process. Calcareous algae are generally far less abundant than corals in the Red Sea and our data does not suggest that this situation is changing (though only field surveys can determine this).

Line 140: Your discussion talks extensively about dissolution, which is assumed here to remain constant. Please revise or clarify.

Thank you for pointing this out. Clarification was added in the text “In-situ dissolution of CaCO_3 at the site of its formation, as part of the diurnal cycle, does not measurably modify the chemistry of the water far away from the reef as long as dissolution is balanced by accretion, hence this natural process is normally transparent in our calculations.”

Line 167: Sentence is very confusing. Please revise.

The sentence was reworded to “The IO A_T -to-salinity slope in 1998 and 2015-2018 remained unchanged with a value of 61.9 ± 4.0 .”

Line 222: In which direction are they temperature sensitive? This is very important for the argument, particularly given that temperature played such a large role in this particular event.

The sentence was reworded “It was previously shown that dissolution rates of silicate radiolarians in the undersaturated water column increase exponentially with increasing temperatures (50) and it is likely that the same is true for the far more soluble acantharia shells.”

Line 263: To what degree does this change increase the saturation state?

The sentence “At the temperature-salinity range in the study area, increased temperatures increase the aragonite saturation state by 0.03 units per degree Celsius and may either increase or decrease biotic calcification rates, depending on the physiological response of organisms to temperatures.” was added to the text.

What were the actual temperature differences between 1998 and 2015? It seems this is extremely important for the claims made below. If warming was greater, wouldn't this have a larger impact on saturation? Bleaching? Was this warming the actual observed differences during one of the worst bleaching years on record? Generally, more reporting of actual measured values would be helpful.

Thank you for the comment. Sea surface temperatures measured on the ship during the December 2015 – January 2016 sampling have been added to the supplementary material table. The winter time temperatures are within the normal temperature range of the Red Sea. As referenced in the text, the value we used in our calculations (warming of 0.25°C since 1998) relies on satellite based measurement as calculated and summarized by Chaidez et al., 2017. Their calculations show that 2015 was 0.5°C warmer than average which translates to +0.015 in Ω aragonite. This was not the warmest year on the recent record of this region yet coral bleaching was exceptionally high. Because the change can't be explained by temperature, we have used the text to explore other mechanisms responsible for reduced Red Sea coral calcification rates in recent years.

266: Many different claims are made about saturation state, but saturation was not, to my knowledge, directly measured. A brief explanation of how these numbers were derived specifically for this paper would be helpful, or if they were not measured, a comment on how saturation may have differed between the values measured in the previous paper and the current one?

We thank the reviewer for this comment. The calculated saturation state of aragonite along the Red Sea in 1998 and 2015 was plotted alongside the calculated net community calcification rates calculated for the null hypothesis that change in coral calcification rates was only due to warming and acidification. This new figure was added to the supplemental information:

Figure S1: Calculated aragonite saturation state of Red Sea surface water (left) and calculated net coral reef community CaCO_3 deposition rates (G_i ; right panel) in 1998 and 2015, assuming it had changed only due to acidification and warming. Dashed line at $\Omega=1$ (left panel) marks saturation with respect to aragonite. The calculations assume equilibrium of Red Sea surface water with atmospheric CO_2 pressure as measured at Mauna Loa (ftp://aftp.cmdl.noaa.gov/products/trends/co2/co2_mm_mlo.txt) and warming of Red Sea surface water by 0.25°C during 1998-2015 (1). Calculations were done using the stoichiometric solubility product for aragonite in seawater from (2) and the Eilat reef equation (3, 4). Carbonate ion concentrations were calculated using total alkalinity and pCO_2 by $\text{CO}_2\text{sys v2.1}$ (5) with K_1 , K_2 from (6) and total boron concentrations from (7). Calcium concentrations were directly measured in the present study for the 2015 data and by (8) for the 1998 data.

267: It was stated previously that there were large changes in coral cover due to the bleaching. Please clarify.

Thank you for this. In the revised manuscript, part of the sentence was incorporated in the previous sentence while the rest was removed from the text. The combined sentence is: “A null hypothesis of no change in live coral cover suggests that the increase in pCO_2 and warming of 0.25°C (63) between 1998 and 2015 should have decreased net coral calcification rates along the Red Sea by 4.4%, based on the observed response of a coral reef community in the Gulf of Aqaba to seasonal changes in the saturation state of aragonite and temperature (11; *supplementary Fig. S1*). Therefore, the dramatic decline in net coral calcification rates in the southern and central Red Sea as inferred from Figs. 2-4 must result from additional processes.”

Line 292-295: This sentence is very confusing. Please revise and clarify. As written, the logic is hard to follow and put into context with the argument and previous values reported.

The sentence was removed.

Line 330: These are not standard operating procedures for carbonate system parameters.

Thank you for this but we respectfully disagree. Page 33 in the book “Guide to best practices for ocean acidification research and data reporting”, 2010 (editors Riesbell, Fabry, Hansson and Gattuso) suggests four different methods as best practice in the analyses of total dissolved inorganic carbon and three methods as best practice in the analyses of total alkalinity (see Table 1.3 in chapter 1 “The carbon dioxide system in seawater: equilibrium chemistry and

measurements” by Andrew G. Dickson). The method we used for analyses of total dissolved inorganic carbon is method C from that list: Acidification/ gas stripping/ infrared detection. The method used for determination of total alkalinity is method F from that list: Open-cell acidimetric titration.

Figure 3: This figure, with the labeling as is, is very hard to read. For the color blind, it is nearly impossible to read. Please consider changing the figure to make the points clearer and the overall conclusions drawn more visible.

Thank you for pointing this out. Figure 3 was redrawn to include more data and make it clearer. The new Figure 3 is:

Reviewer #3:

The main objection I have for this paper, is that the authors present two data points, one in 1998 and another in 2015, and attribute the changes in NCC to changes in environmental conditions (temperature, CO₂, and eutrophication). This is problematic for many reasons. First, they do not consider any interannual variability. It would not be surprising to me if planktonic calcification rates exhibited interannual variability of ~7 %, which is the signal they observed here.

Thank you for this comment that Reviewer 2 also pointed out. In the revised submission we added new data from March 2018 and plotted in figures 2 & 3 data from April 2016 that was included in the supplementary information but not plotted before. Inclusion of the new data reveals that there is no significant seasonal variability in alkalinity or strontium-to-alkalinity ratios.

Second, unfortunately 2015 was the year where coral reefs experienced mass bleaching events worldwide, including the Red Sea. A significant decline in NCC would be expected associated with this event. However, the authors make no attempt to quantify how this bleaching event may have contributed to the decline in the observed coral reef NCC. It is plausible that the signal was dominated by this event.

We agree with the claim that coral bleaching reported by scientists from KAUST in the central Red Sea likely reduced coral calcification rates. The paper reports the expected change in calcification rates based on the effects of warming and ocean acidification, and one can assume that the rest is due to metabolic reasons which may be directly related to bleaching. The March 2018 data added in the revised manuscript suggests that coral reefs in the Red Sea have not recovered from the bleaching event or other stress factors even though the global bleaching event had ended by 2018.

The Eilat reef equations assumes that there is no change in community composition, however, this is not a valid assumption. The authors mention that there has been documented declines in coral cover in this region. In addition, there was a massive global bleaching event in 2015 that affected reefs worldwide, including the Red Sea. Therefore, it is hard to believe any inference based on this equation, and I recommend it to be removed from the manuscript.

We thank the reviewer for scrutinizing our working hypothesis. The Eilat reef equation serves as the null hypothesis that Red Sea calcification rates changed only due to changes in the saturation state of the mineral aragonite. The observation that coral calcification rates decreased in great excess of predictions made using the saturation state of aragonite suggests that we are observing changes in the community structure of Red Sea corals or metabolic state of the coral reefs (or likely both). Additional observations that coral calcification rates were still low in 2018 support this conclusion. It is thus apparent that the null hypothesis of no change in the state of the Red Sea reefs embedded in use of the Eilat reef equation is incorrect, leading us to adopt an alternative hypothesis that the metabolic state of the reefs and their community structure were altered. Due to this reason, we agree with the reviewer that the assumptions embedded in the Eilat reef equation are violated but think that it is important to leave this discussion in the text. To avoid confusion for other readers, reference to this equation was removed from the abstract but maintained in the body of the manuscript.

The authors mention that the Rayleigh distillation model does not work if dissolution is a significant process, however, later goes on to argue that dissolution rates in coral reefs increased by 75% (from 20%). They do not mention how this would affect the model calculation (I assume another end member needs to be added for reef dissolution?) An uncertainty analysis for this seems warranted.

We thank the reviewer for pointing out this unclarity. The Rayleigh model only tests net precipitation of CaCO₃. It is very sensitive to the occurrence of net dissolution but will only be affected by gross dissolution if precipitation and dissolution are significantly offset in space and/ or time. This point was clarified in lines 143-146 “In-situ dissolution of CaCO₃ at the site of its formation, as part of the diurnal cycle, does not measurably modify the chemistry of the water far away from the reef as long as dissolution is balanced by accretion, hence this natural process is normally transparent in our calculations.”

Finally, how can you be confident that the Central and Southern part of the Red Sea is unaffected by deep sea processes? If it is altered in the North, I would assume it would also have an influence in the Central and Southern parts as well. If deep waters have a different Sr/TA, Sr/Ca ratio, then this could affect the interpretation.

The Red Sea deep water has salinity of ~40.7 and temperature of 21°C. South of 23°N the salinity is at least a unit lower (and decreasing southward) while temperatures do not normally go below 25°C in winter. The density gradient between the surface and the intermediate to deep waters is therefore fairly steep at the south and central parts of the Red Sea basin. This point was made clearer in the text (Lines 100-102) “Upper thermocline water in the south and central Red Sea is warmer throughout the year and less salty than the deep and intermediate waters of the Red Sea, suppressing deep water formation and ventilation in these regions.” And lines 193-196 “Unlike the northern Red Sea, mixing of brine waters into surface waters of the central and southern parts of the Red Sea is of minor importance due to the development of a steep density gradient between the surface and deep waters.”

Specific

L38: A follow up paper to this experiment has now been published in Nature. So there are now 2 controlled field experiments.

A reference to the newly published paper by Albright and colleagues was added.

L146-149: A single entry point into the Red Sea is not a good justification of a laterally mixed system.

Clarification of this claim was added to the text (lines 151-153): “Within this channel, a series of mesoscale eddies vigorously mixes the upper water column on the longitudinal axis (28), as reflected by the zonation of many physical and biological parameters (38).”

L266: This assumption is not true, due to the changing coral community mentioned by the authors throughout the 2000’s, and the bleaching event in 2015.

We agree with the reviewer and reworded the sentence to reflect this: “A null hypothesis of no change in live coral cover suggests that the increase in pCO₂ and warming of 0.25°C (63) between 1998 and 2015 should have decreased net coral calcification rates along the Red Sea by 4.4%, based on the observed response of a coral reef community in the Gulf of Aqaba to seasonal changes in the saturation state of aragonite and temperature (11; *supplementary Fig. S1*). Therefore, the dramatic decline in net coral calcification rates in the southern and central Red Sea as inferred from Figs. 2-4 must result from additional processes.”

L274-L281: It is not clear to me what the authors are arguing here.

Parts of the paragraph were reworded “Reduced rates of coral calcification and reduced abundance of large corals since 1998 have been previously reported for several Red Sea reefs (14, 60). It was suggested that this decline in coral growth rates was a result of a long series of warm years in the central Red Sea (14). In 2015, global temperatures were particularly high and widespread coral bleaching events were reported globally, including sites in the central Red Sea, Persian Gulf and Indian Ocean (18, 50, 62). In the Red Sea, sea surface temperatures in 2015 were 0.5°C higher than the long term average, yet while 2015 was warmer than average for this region, it was not the warmest on record in recent years (62). The bleaching event clearly affected the metabolic state of Red Sea corals yet coral bleaching is a very rapid process, which is unlikely to induce a gradual decline in coral growth as documented for the central Red Sea (14). It therefore seems that while bleaching played a major role in decreasing coral calcification rates, change started earlier. Data from April 2016 and March 2018 show that recovery from the 2015 bleaching event and contributing stress factors is yet to happen.”

L292-L297: These calculations are dubious. See comments above.

These sentences were removed.

L307: the strontium budget suggests that their calcification rates may have decreased, but doesn't necessarily mean their abundance has decreased.

The word “abundance” was replaced by “calcification rates”.

L316: How do you get footprint of the chemical data without information of circulation patterns?

Our calculations take the implicit assumption that the hydrography of the Red Sea did not change considerably in recent years. We see no reason to question this assumption after reviewing the recent literature on the hydrography of this region.

L330: Were the samples poisoned? It sounds like the samples could have sat around for 2 months before they were analyzed. If it wasn't fixed properly, then TA could get altered.

DIC samples were poisoned upon subsampling and total alkalinity samples were measured within a day or two after samples arrived in Israel.

L343: a commercially available system isn't exactly 'custom'...

The word “custom” was deleted.

Figure 3: Is this all of the data? Or just south or 20 N?

The legend of Figure 3 was reworded to clarify that the figure presents all the data “Total alkalinity versus salinity in all surface water samples collected from the Indian Ocean (IO), Arabian Sea, Gulf of Aden and Red Sea (RS) in November 1998, December 2015, April 2016 and March 2018. Analytical uncertainties are smaller than symbol sizes.”

We thank the reviewers for their helpful comments and suggestions.

REVIEWERS' COMMENTS:

Reviewer #2 (Remarks to the Author):

While I still think caution should be taken on the overall claims based on the severity of the bleaching years in question (original comments mirrored by another reviewer), the authors have responded to all of my previous comments satisfactorily.

Reviewer #3 (Remarks to the Author):

The addition of the new data sets is a great addition to the manuscript, and addresses my major concerns with this paper. The modification of discussion for the Eilat equations as a null hypothesis is a welcome addition as well. The paper is now more convincing, and I recommend publication of this manuscript.

Response to reviewers

Reviewer #2 (Remarks to the Author):

While I still think caution should be taken on the overall claims based on the severity of the bleaching years in question (original comments mirrored by another reviewer), the authors have responded to all of my previous comments satisfactorily.

We thank the reviewer for helping us improve the manuscript through his comments.

We agree with the reviewer that one should expect recovery of the Red Sea coral reefs in the coming years. Before analysing the 2018 data, our prediction was that we would find a significant recovery in Red Sea coral calcification rates relative to 2015 following the end of the global bleaching event. Unfortunately, we cannot find signs of recovery three years after the beginning of the bleaching event. In the Red Sea, the heat wave of 2015 was not the worst on record in the last two decades, and the degree of aragonite saturation is still very suitable for coral calcification. Our conclusion from this is that correlations between water temperature, ocean acidification and coral calcification rates oversimplify the system, while other factors affecting ecosystem health such as eutrophication and overfishing could play important roles in determining the rate of coral reef community calcification and CaCO_3 dissolution.

Reviewer #3 (Remarks to the Author):

The addition of the new data sets is a great addition to the manuscript, and addresses my major concerns with this paper. The modification of discussion for the Eilat equations as a null hypothesis is a welcome addition as well. The paper is now more convincing, and I recommend publication of this manuscript.

We thank the reviewer for helping us improve the manuscript through his comments.